# Preliminary Statistical Characterizations of the Lowest Kilometer Time–Height Profiles of Rainfall Rate Using a Vertically Pointing Radar

**Arthur R. Jameson** [1,*] and **Michael L. Larsen** [2]

1    RJH Scientific, Inc., El Cajon, CA 92020, USA
2    Department of Physics, College of Charleston, Charleston, SC 29424, USA; larsenml@cofc.edu
*    Correspondence: arjatrjhsci@gmail.com

**Abstract:** A realistic approach for gathering high-resolution observations of the rainfall rate, $R$, in the vertical plane is to use data from vertically pointing Doppler radars. After accounting for the vertical air velocity and attenuation, it is possible to determine the fine, spatially resolved drop size spectra and to calculate $R$ for further statistical analyses. The first such results in a vertical plane are reported here. Specifically, we present results using MRR-Pro Doppler radar observations at resolutions of ten meters in height over the lowest 1.28 km, as well as ten seconds in time, over four sets of observations using two different radars at different locations. Both the correlation functions and power spectra are useful for translating observations and numerical model outputs of $R$ from one scale down to other scales that may be more appropriate for particular applications, such as flood warnings and soil erosion, for example. However, it was found in all cases that, while locally applicable radial power spectra could be calculated, because of statistical heterogeneity most of the power spectra lost all generality, and proper correlation functions could not be computed in general except for one 17-min interval. Nevertheless, these results are still useful since they can be combined to develop catalogs of power spectra over different meteorological conditions and in different climatological settings and locations. Furthermore, even with the limitations of these data, this approach is being used to gain a deeper understanding of rainfall to be reported in a forthcoming paper.

**Keywords:** raindrop size distributions (DSDs) from Doppler radar; computing radial power spectra using radar Doppler spectra; vertically pointing Doppler rain observations

## 1. Introduction

The correlation functions and power fits of spectral powers have been used extensively to relate measurements of rainfall rates at different temporal and spatial scales (e.g., [1–5] and many others). However, such studies have concentrated exclusively on horizontal dimensions and time because of the difficulty of measuring rainfall rate in the vertical dimension, especially at high resolutions, over any significant depth. This study is a first step toward addressing this deficiency.

This is achieved by using vertical observations of rain using a Micro-Rain Radar (MRR), which is a continuous-wave Doppler radar operating at a wavelength of 1.24 cm, as described in detail in [6]. It has selectable vertical resolutions, integration times, and sampling intervals. In this study, we use a vertical resolution of 10 m over a depth of 1.28 km with 10 s integration and 64-point Doppler spectra over an unambiguous Doppler velocity range of approximately 12 m s$^{-1}$.

These measurements are affected by both attenuation by the rain and vertical air velocity, which can distort the raindrop size spectra and the estimated drop concentrations used to calculate rainfall rates and other parameters. Both effects can be taken into account and corrected, as described in detail in [7]. The lengthy discussions therein will not be repeated here except to say that the approach uses velocity-shifted Doppler spectra until

the observed and theoretical powers agree. Furthermore, we emphasize that this is not a work about what has been done in the past using MRRs particularly in snow, for example, or how to use these radars, but rather the focus here is on an interesting first atmospheric application in rain that will be expanded upon further in a future study already under preparation and that will be described in greater detail at the end of this work. As such, this work should only be considered as the presentation of some analysis techniques and the results as preliminary, with no broad generality at this point.

However, there is no guarantee that the tools of proper correlation functions and power laws will always exist. In particular, the correlation function exists only when they (mean values and variance) are independent of the origin of their calculation in space or time over the spatial–temporal domain of interest. Similarly, power spectra, whether power fits or otherwise, exist but only have generality when the data are widespread, statistically homogeneous, and statistically stationary (WSS), as emphasized for the rainfall rate in [8]. According to the Wiener–Khintchine theorem, only when the data are WSS can one compute the auto-correlation function and proper power laws [1,2]

Specifically, then, the first order of business is to see whether or not the temporal–vertical MRR observations of rainfall rates are statistically homogeneous. There are two components to this determination. First, at all times and in all directions, there has to be only one global mean value. Second, the variance must be the same at all times and in all directions as well. In order to address the first requirement, a method of inverting individual observations is used to look at the distributions of the mean values ([8–11]). When there is a unitary peak in the resulting distribution, this condition is satisfied. It should be noted, however, that while an entire dataset may not satisfy these conditions, they may be locally satisfied. Whether or not these local regions are useful remains to be seen.

The variance requirement is addressed using the results of Anderson and Kostinski ([12,13] through the analysis of the difference in the number of sequential maxima and minima forward and backward directions in a string of data denoted by the variable $\alpha = T_{\text{foreward}} - T_{\text{backward}}$ where $T$ is the total count of record highs and record lows in each direction. For a sample size greater than ten, $\alpha$ is normally distributed with a null mean and a standard deviation $\sigma_\alpha$ dependent on the sample size (Figure 1 in [14]). Asymmetries in the variance appear as a non-zero mean $\alpha$ of a magnitude that can then be statistically evaluated with respect to $\sigma_\alpha$. Examples of these applications of MRR data are provided in the next section with the analysis results for four different sets of data provided subsequently.

While some argue that these two requirements are 'too restrictive' for real rain, these, unfortunately, are the mathematical requirements for WSS. Furthermore, because the data analyzed here are along two orthogonal dimensions, one must apply both criteria in both directions to evaluate the appropriateness of WSS over the area.

## 2. Examples of the Data Processing for Determining Statistical Homogeneity for Time–Height Rainfall Rate Data

### 2.1. Convective, Variable Rain

Figure 1 is a plot of the time–height MRR vertical air velocity, attenuation-corrected rainfall rates using observations that were collected by an MRR radar as part of a National Science Foundation project and operated by the College of Charleston, located near Charleston, South Carolina. It is located on property owned by the College of Charleston Foundation that is used for a variety of ecological, educational, and research purposes (e.g., see [15]). The methodology for the correction of the data for vertical air velocity and attenuation was as explained in [7]. As one would expect for this time of year in South Carolina, the rain was associated with convection, having a wide range of rainfall rates. The most noticeable feature overall was the vertical structure of the rain that, of course, was not surprising in convective rain.

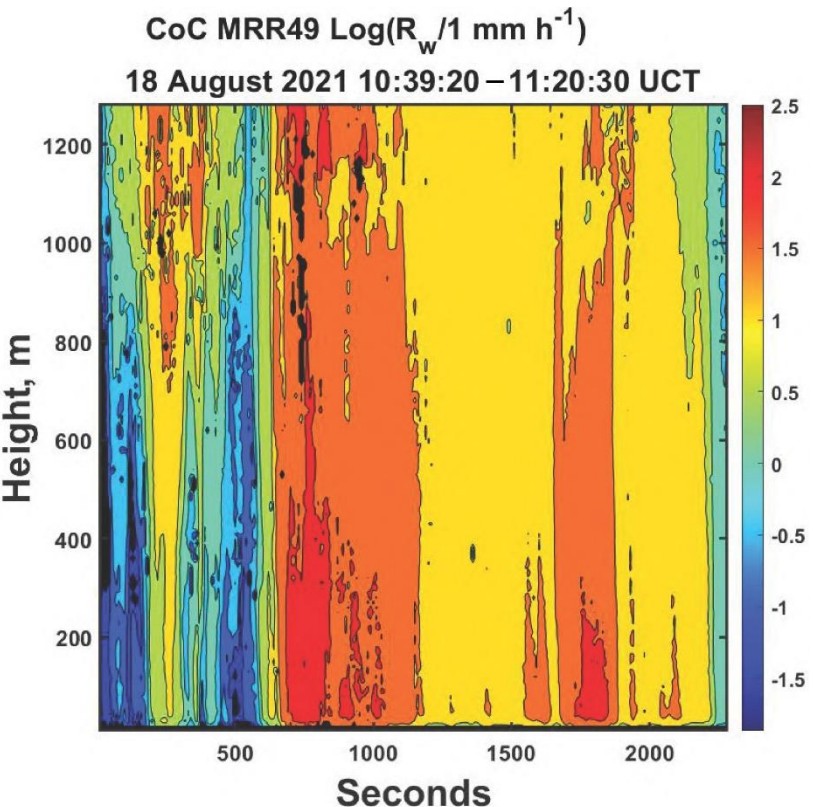

**Figure 1.** Time–height plot of the base 10 logarithm of the vertical-air-velocity-corrected Doppler spectra rainfall rate from one of the College of Charleston MRR radars illustrating rain shafts typical of summer convective rain there. Areas of black denote missing data.

However, it did suggest that the determination of statistical homogeneity has a strong dependency on direction. Since truly statistically homogeneous data is independent of direction of measurement, this means that any regions of such data would be those locations where the calculations of statistical homogeneity in both the vertical and the horizontal directions overlapped.

Specifically, to explore each direction, an optimal way of looking at the two-dimensional data was to stack each column to form two one-dimensional arrays, one for time and one for the vertical dimension, as discussed on p. 1406 in [14]. A local regression mean curve (a least square error fit over twice the decorrelation length) was then fit to these data, and the deviations from the mean gave the fluctuations used in the subsequent analyses using $\alpha$. The results are illustrated in Figure 2a for the temporal dimension and in Figure 2b for the vertical direction. Obviously, there were significant differences between the two. The grey horizontal lines largely correspond to symmetrical regions in the fluctuations so that $\alpha$ should be relatively constant. These regions were then used in the analyses. The blue line in Figure 2b highlights the ever-changing values of the fluctuations so that $\alpha$ would also be ever-changing. Hence, the entire region was selected as one block of data, as indicated by the extended grey line.

The more symmetric region of fluctuations is identified by the second grey line. In each of these sections, the numbers of contributing mean value components ($N_b$) were determined by the number of peaks in the posterior frequency distribution of the mean $R$ from Bayesian analysis of the observed rainfall rates. In addition, the $\alpha$ analyses were performed separately to yield the $\alpha$ relative dispersion $RD_\alpha = |\alpha| / \sigma_\alpha$ where $|\alpha|$ is the absolute value of $\alpha$, and $\sigma_\alpha$ is the standard deviation of $\alpha$ given by

$$\sigma_\alpha = [4ln(n) - 4.271]^{1/2} \tag{1}$$

where $n$ is the number of measurements in the string of observations (from [14]). In that same article, an index of statistical homogeneity was then defined to be the combination of these two factors:

$$IXH = \frac{1}{2}\left[H\left(\frac{RD_\alpha}{1.5} - 1\right)\left(\frac{RD_\alpha}{1.5} - 1\right) + (N_b - 1)\right] \tag{2}$$

where $H$ is the Heaviside unit step function requiring $RD_\alpha$ to exceed 1.5. We referred that term to the alpha factor, and the second term was the number of mean values (Bayesian) factor. In purely statistically homogeneous data, $\alpha = 0$ and $N_b = 1$ so that $IXH = 0$. In reality, these are very restrictive conditions rarely seen in real data, so we used $IXH \leq 0.5$ as a sufficient indication of statistical homogeneity.

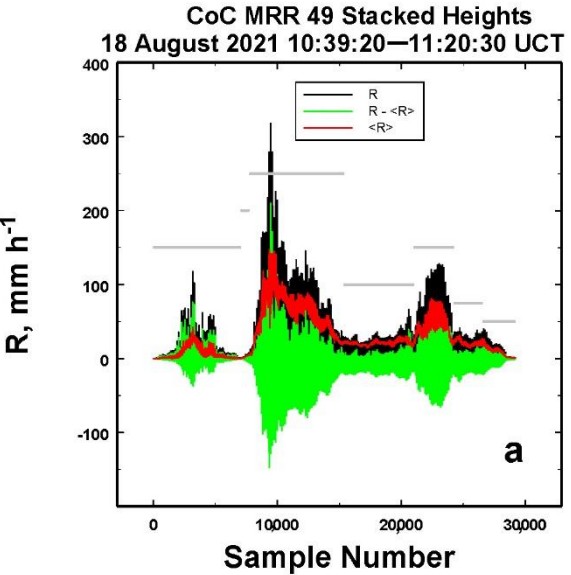

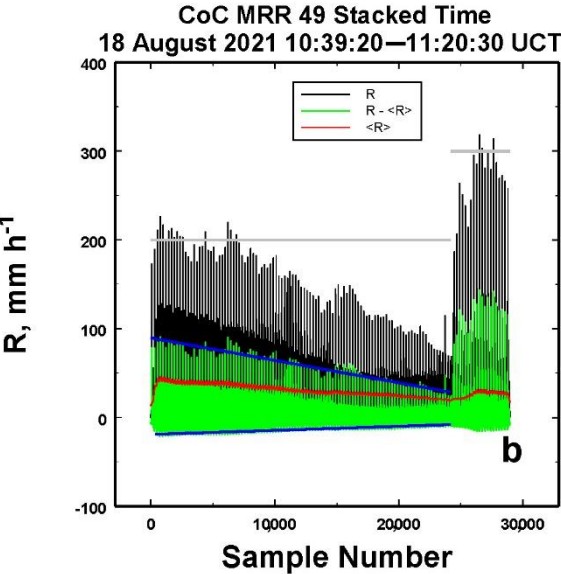

**Figure 2.** Data series constructed from: (**a**) stacking the sequential observations of height in Figure 1 into a single vector (black lines) with the computed mean curve (red) and fluctuations (green) and (**b**) from stacking sequential time observations over all time. As explained in the text, the grey lines denote the breakdown into regions for subsequent calculations of the statistical homogeneity *IXH*. In (**b**), the blue line indicates where the fluctuations are constantly changing.

In truly statistically homogeneous conditions, all results independently in each direction should independently be the same in each direction. Obviously, in general, that was not the case for these data, but it did not rule out local regions where such equivalence may be approximately valid, as suggested by Figure 3, which shows plots of the average values over the combined results across the sample numbers in the temporal and vertical directions. The shaded regions are where the data were statistically homogeneous. By and large, the fluctuations never satisfied the requirement for statistical homogeneity, except beyond about sample number 2500. While there were a few more locations where the mean value factor was satisfactory, it was only beyond about sample number 2600 when all the conditions for statistical homogeneity were met.

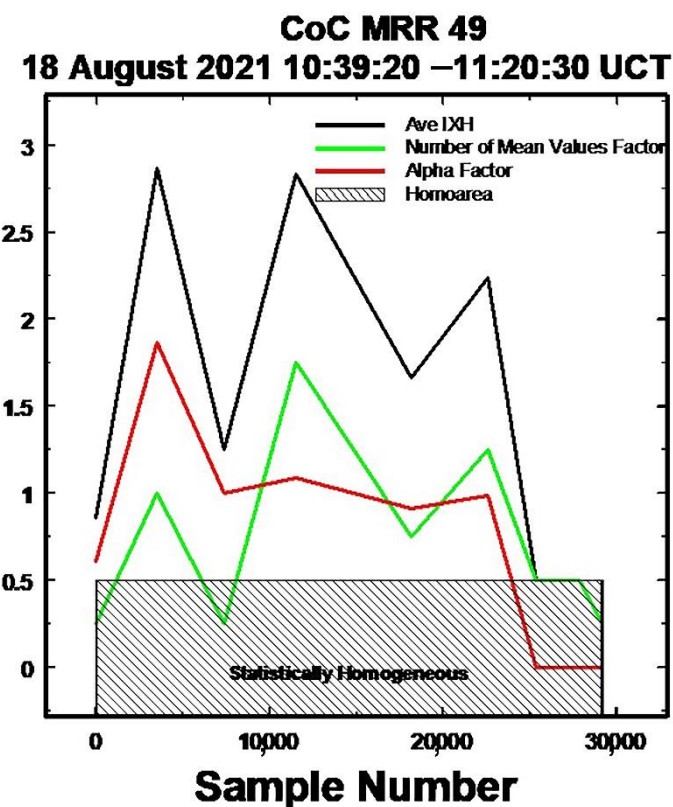

**Figure 3.** Plots of the *IXH* averaged over the temporal and height vector series for each of the grey areas in Figure 2 placed at the mid-points of each region. Only at the right-most part of the figure is there any sign of statistical homogeneity.

To see where in space and time these conditions were met, we first interpolated the *IXH* values in the space series and in the time series separately. These were then unstacked to return them to their original time–height locations, and, finally, these were then averaged together to estimate a combined field. We then imposed two requirements for statistical homogeneity on the resulting field of data. The first was that $IXH \leq 0.5$, and the second was that, in locations satisfying this first requirement, the absolute value of the difference between the two fields for each direction separately was $\leq 0.3$. This latter requirement was designed to satisfy the directional independence of statistical homogeneity.

The results are illustrated in Figure 4, where the contours of shading indicating where statistical homogeneity was possible (brighter areas) and where it was not (darker areas) are overlaid on the rainfall rates. The first obvious feature is that, with the exception of a tiny narrow region at the top-left, these data were all statistically heterogeneous.

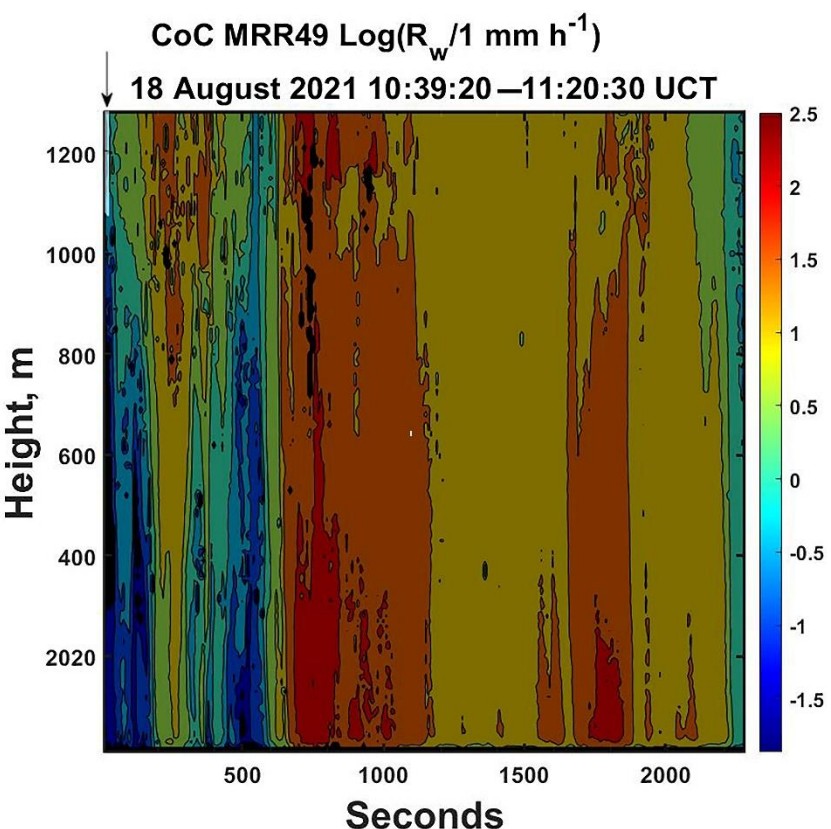

**Figure 4.** A replot of Figure 1 with an overlay of the height and time 2D *IXH*. Lighter areas denote where there is statistical homogeneity subject to the two requirements discussed in the text, while the darker areas denote where there is only statistical heterogeneity. For this set of observations, there is only one narrow region of statistically homogeneous data in the top-left, indicated by the arrow.

To see whether these results also applied to other data, we next considered three more sets of data (all from 3 June 2019) measured using a NASA MRR-Pro radar located at the Wallop's Island Flight Facility. The rainfall rates were already previously determined, as explained in [7]).

These observations were broken into three segments denoted as early, middle, and later pieces. The rainfall rates and the analysis results for the early period are shown in Figure 5. Because of the profound convective nature of this part of the storm with widely varying rainfall rates over short times, there were no locations of any statistical homogeneity in a manner quite similar to the previous case.

However, even during the middle period of much lighter precipitation, only a few small regions of statistical homogeneity were found at times in the lower few hundred meters, as plotted in Figure 6.

During the later time period, there was a region of light rainfall followed by a period of more intense rain, as shown in Figure 7a. In this later period, there were a few larger regions of statistically homogeneous data, but still, by and large, the rainfall rates remained statistically heterogeneous.

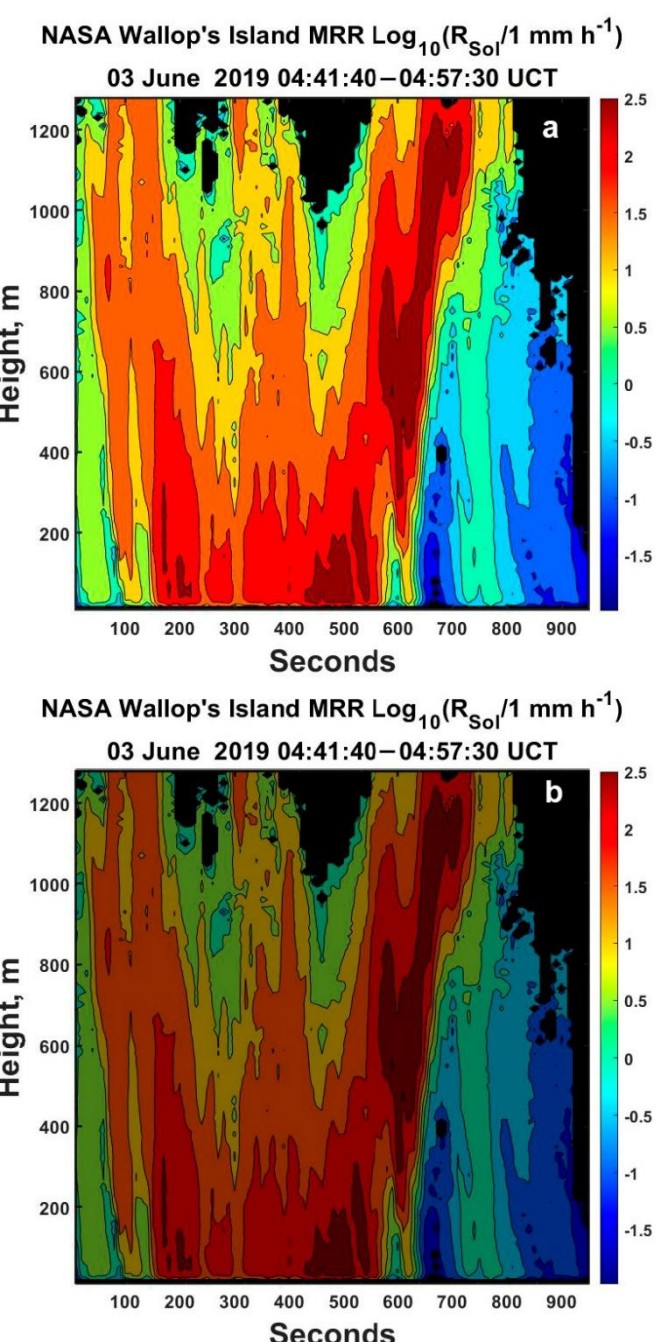

**Figure 5.** (**a**) Time–height plot of the base 10 logarithm of the vertical-air-velocity-corrected Doppler spectra rainfall rate using data from a NASA Wallop's Island MRR radar (from Jameson [7]) during the passage of a line of convection and (**b**) the same plot with an overlay of the 2D *IXH* results. Note that this time there are no regions of statistical homogeneity, apparently because of the variability of $R_w$ in both height and time.

The result was that, for all four of these convective rainfalls, the data must be considered to be statistically heterogenous. This means that correlation functions in time and height did not exist. In so far as these data were representative of typical convective rain, it also seemed plausible that this will be true for most convective rain. What happens in steadier, more stratiform rain remains to be determined.

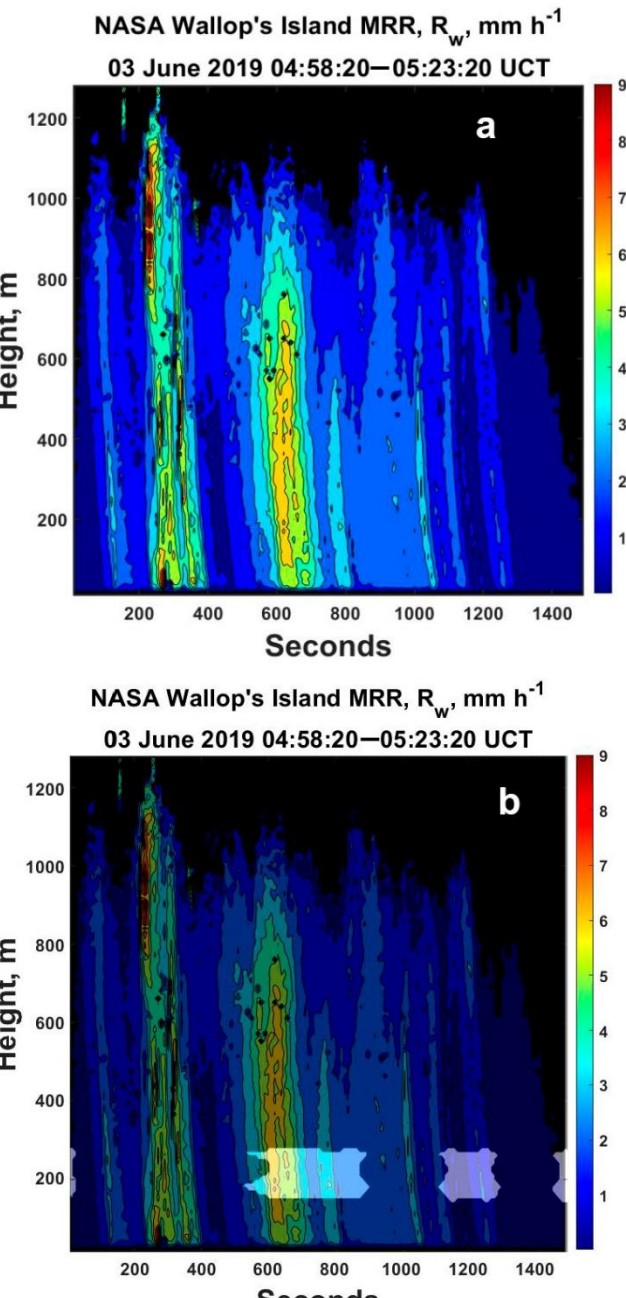

**Figure 6.** Similar to Figure 5 except for in the subsequent 25 min. This time, there are larger areas of statistically homogeneity, but still they are confined and found only at locations below the 300 m height.

Thus, correlation functions are usually likely to be of little use when trying to transform rainfall rates between different time or different spatial scales, nor can they be transformed into power spectra with any general applicability (e.g., see [8]) via the Wiener–Khintchine theorem [16,17]. However, the power spectra of these data fields might still serve a useful, albeit narrower, purpose.

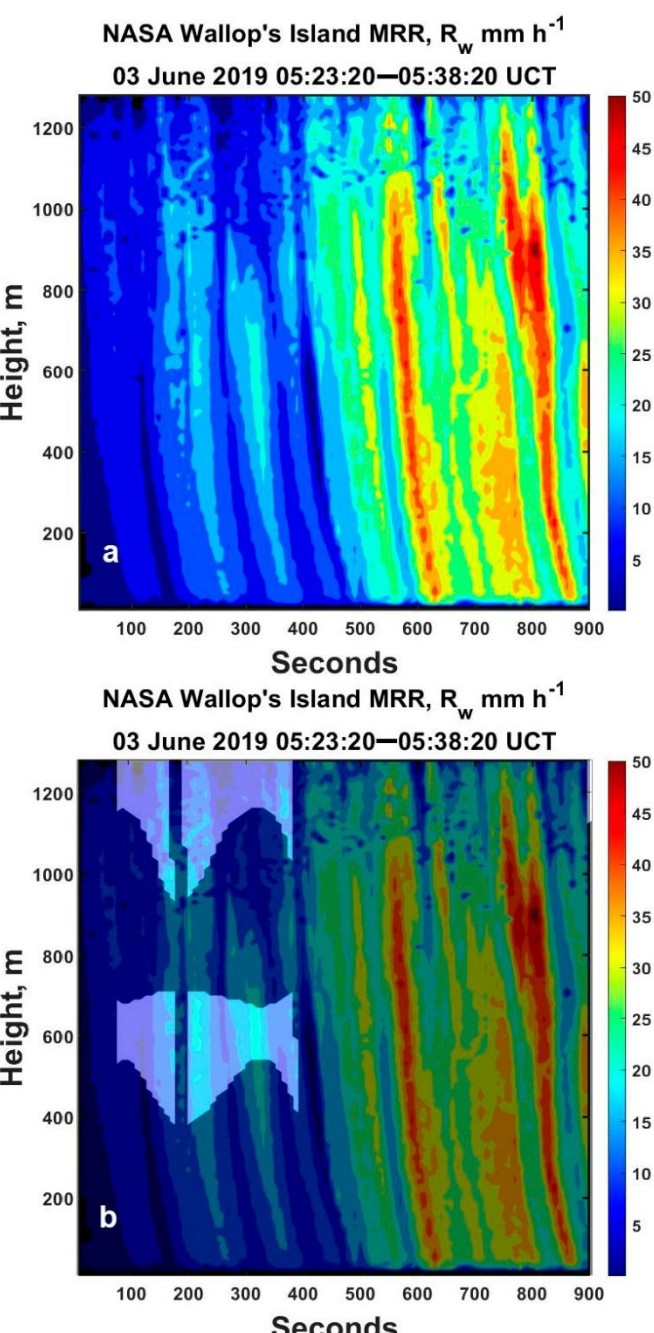

**Figure 7.** Similar to the previous Figures 5 and 6 except for the final sequential 15 min that begin with lighter rain followed by regions where the rainfall is more intense. Some expanded locations of statistically homogeneous rain are found in the locations of less intense rain, but they are still rather isolated in both dimensions (adapted from Jameson [7]).

To explore further, and until we have access to two-dimensional spatial data, the temporal axes in Figures 4–7 were converted into spatial coordinates by assuming an advection velocity of 1 m s$^{-1}$. This yielded horizontal dimensions (i.e., 900–1490 m) for the NASA MRR data and up to 2280 m for the College of Charleston MRR 49 observations, with all having a vertical distance of 1280 m. In each case, the rainfall rate data were then Fourier-processed to yield the two-dimensional power spectra that could then be transformed into the one-dimensional spectra in height and in the horizontal direction (time) for each period. These are illustrated in Figure 8.

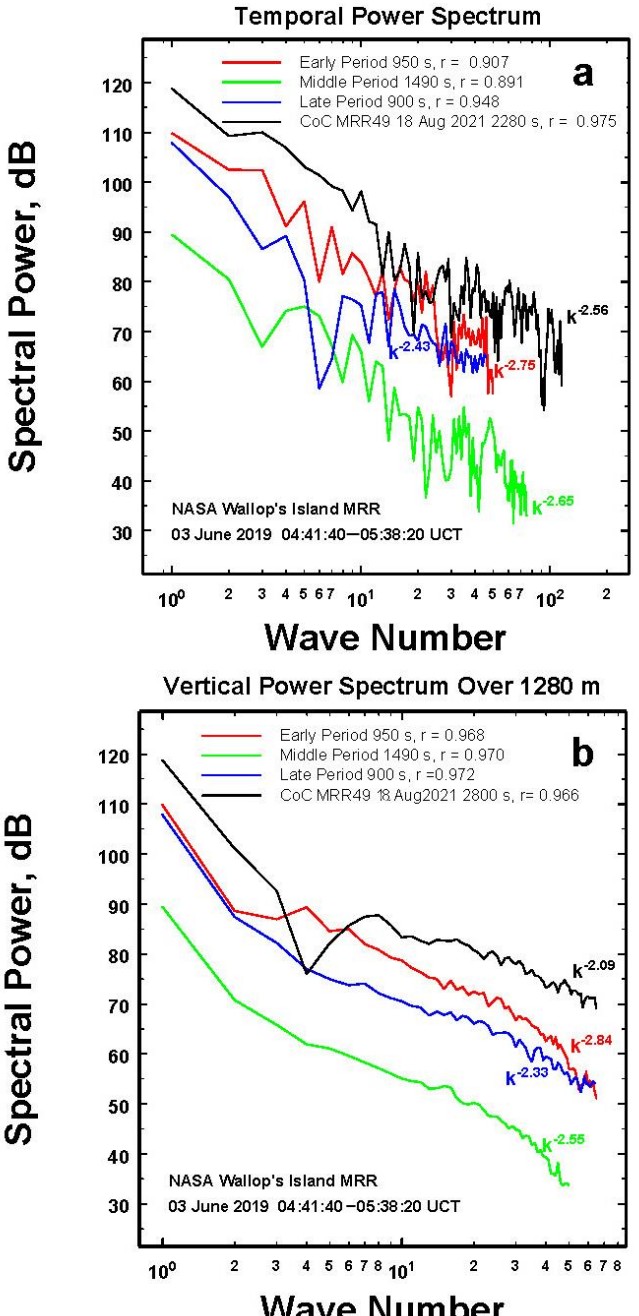

**Figure 8.** One-dimensional power spectra in time (**a**) and height (**b**) with power fits as functions of wave number. In (**a**), the wave numbers for the different times are listed in the legend, while in (**b**), the wavenumbers are per 1280 m.

All of these power spectra can be fit using power functions to a reasonable degree of correlation. Many of the exponents were quite similar, regardless of being in the vertical or in the horizontal (temporal) directions. While the vertical axis covered several orders of magnitude, with the exception of the horizontal power spectrum of the MRR 49 data, the wavenumbers were shy of the two orders of magnitude required for designating them to be a 'power-law' according to the findings of [18]. On the other hand, the general similarity of the various fits suggested that it might be useful to combine the data in the two dimensions.

This was performed next by computing the one-dimensional radial spectra, regardless of time or altitude, as illustrated in Figure 9. This was accomplished first by converting the temporal axis into a spatial dimension assuming a mean advection speed of 1 m s$^{-1}$. The 2D power spectrum in this new horizontal–vertical coordinate system was first computed

by using the fft2 routine in Matlab® and then multiplying by its complex conjugate. This 2D power spectrum of values at ($\Delta x, \Delta y$) was then moved to be those for a 2D polar coordinate system of ($\Delta r, \Delta \theta$) values. The radial spectra were then computed by integration over all the $\Delta \theta$ for each $\Delta r$.

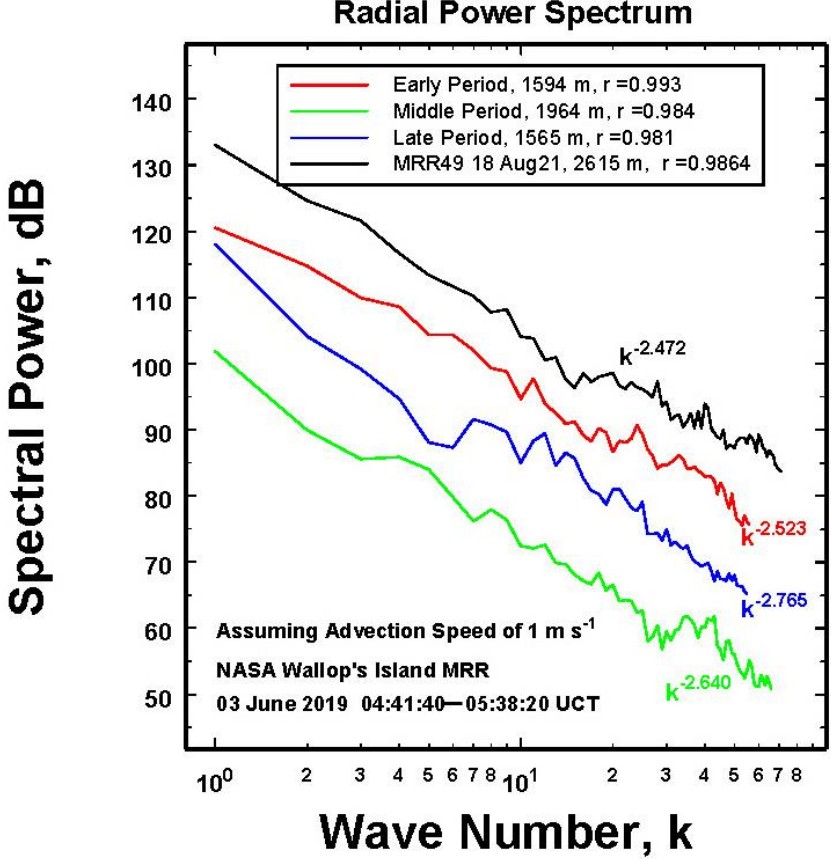

**Figure 9.** The radial power spectra for the data in Figure 8 with *k* being the wave number for the distances indicated in the legend.

The intercept at $k = 1$ provided a good measure of the total variability of the data at the different times. All of the slopes were quite similar. However, these values were consistently larger in magnitude than many reported in the literature, which usually range from 1–2 for large horizontal and temporal domains. As [19] pointed out and Molini et al. [20] reemphasized, the magnitude of the exponent increases as the time and space scales decrease. For the data here, then, it was not surprising to see larger exponents because the temporal and spatial domains of these measurements were smaller and finer compared to what has normally been used. In addition, no other studies have been able to look at the vertical plane in this detail, complicating any comparisons to previous observations. Consequently, we took these observed slopes at face value within the restrictions presented.

However, exact magnitude also depends upon the assumed average advection velocity, as illustrated in Figure 10. When the advection velocity increased to 5 m s$^{-1}$, the negative slope increased significantly in magnitude.

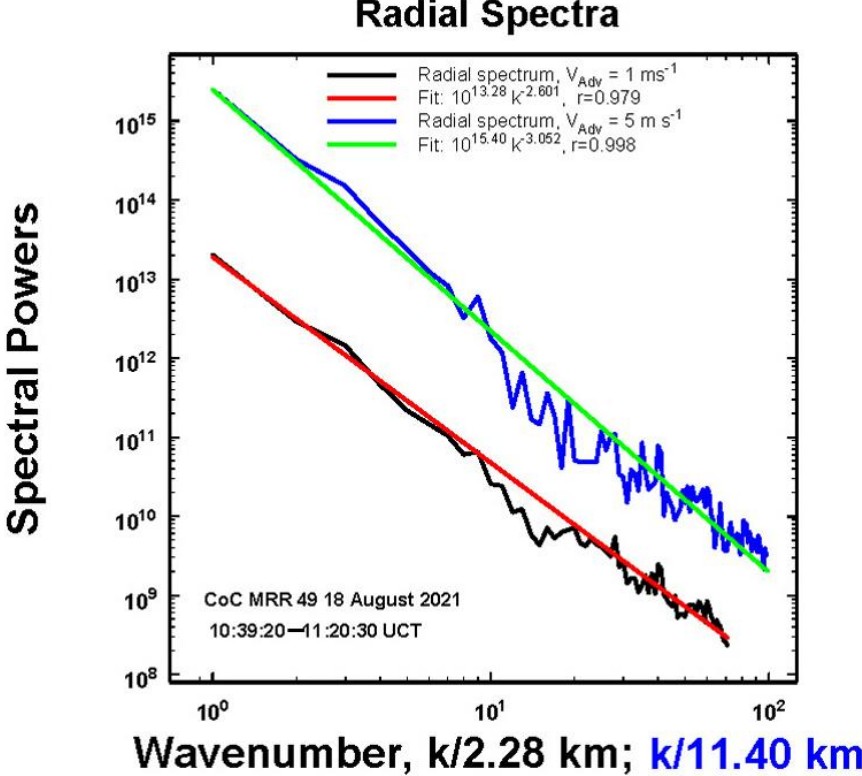

**Figure 10.** An example of the radial power spectra corresponding to two different assumed advection velocities. The increase in the magnitude of the negative slope is discussed in the text.

To understand this, consider a particular Fourier wavelength component describing the rain field for $1 \text{ m s}^{-1}$ advection velocity. When this wavelength is instead moving at $5 \text{ m s}^{-1}$, the wavelength is 'stretched' compared to what it was at the $1 \text{ m s}^{-1}$ velocity. To put it another way, the wavelength increases by a factor of 5 so that the wave number is decreased by a factor of 5. This means that more and more of the spectral energy is moved from shorter to longer wave lengths so that the radial power spectrum now shows a steeper tilt (negative slope).

This, then, highlights the limitation of trying to convert a time–height profile into representative 2-D spatial data so that, ultimately, the statistical analysis of 2-D height–distance rainfall data must be based upon using direct simultaneous observations by a line of several vertically pointing radars. As a first step toward this goal and part of current funding, we will collect simultaneous measurements using two MRR-Pro radars, but because of extenuating circumstances, we have yet to gather such data.

### 2.2. Lighter, Steadier Rain

Figure 11 is a plot of the rainfall rate in a winter rainstorm at Wallop's Island, Virginia. Obviously, the rainfall rates during this period were less intense rain than in the previous sets of analyzed data above. For these observations, the peak frequency of occurrence was $2.7 \text{ mm h}^{-1}$, and the mean rate was $4.7 \text{ mm h}^{-1}$ with a few embedded regions of more intense rain. Over the entire period, calculations showed that *IXH* = 1.0, so the data as a whole were statistically heterogeneous. However, within these data, there was a period of lighter, apparently steadier rain (i.e., a nearly constant mean *R*).

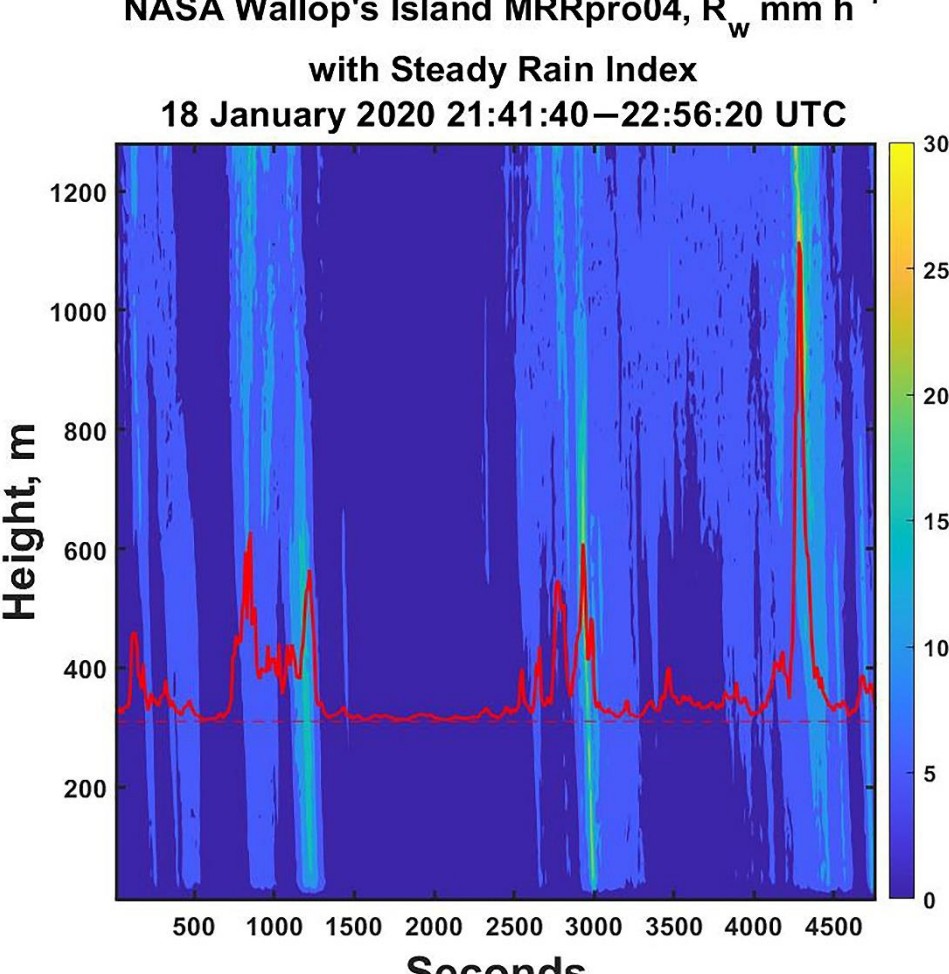

**Figure 11.** The air-velocity-corrected rainfall rate during a winter rain event. The red line is the calculated steady rain index, and the dashed line denotes perfectly steady rain.

To see whether or not the rain in the region from 1340–2390 s was truly steady, we used the approach of Jameson and Kostinski [21] to define a steady rain index (*SRIndx*) using their Equations (8) and (11):

$$SRIndx = \frac{\sigma_R^2}{\sigma_P^2} = 1 + \overline{R}^2 \left[ \frac{\sigma_n^2}{\overline{n}^2} - \frac{1}{\overline{n}} \right] \tag{3}$$

where $\sigma_P^2$ is the variance of the rainfall rate with a Poisson distribution of the total number of raindrops *n* during the observation and a mean rainfall rate *R* equal to the observed mean rainfall rate; $\sigma_R^2$ is the variance of the observed rainfall rate during the observation; $\sigma_n^2$ is the variance of the observed number of drops during the observation; and $\sigma_n^2 = n$ is the observed mean number of drops.

There are two ways to calculate these latter quantities. One is to look at the data for height at each time, and the other is to look across all times at a particular height. It is the former method that made sense here. The *SRIndx* is plotted as the solid red line in Figure 11. When the rain was steady, the number of drops was Poisson [21], and the *SRIndx* = 1 because, for Poisson rain, $\sigma_n^2 = n$. This is indicated by the dashed line in Figure 11.

There was only one 17.5 min period when the rain could be considered to be very steady (between 1340 to 2390 s) when the solid rain line is very near to the dashed line. In that location, the statistical homogeneity index was found to be 0.10 as well, so these

observations could also be considered statistically homogeneous, allowing the calculation of a correlation function and more general radial power law.

This is illustrated in Figure 12, where an equivalent-length radial spectrum for neighboring statistically heterogeneous data is also plotted. Obviously, in this instance there was a noticeable difference between the two radial spectra, emphasizing that one cannot simply use statistically heterogeneous data as a substitute for statistically homogeneous data, and visa versa. For completeness, from the fit in Figure 12, the corresponding correlation function for the homogeneous data was $\rho(x) = x^{-0.200}$.

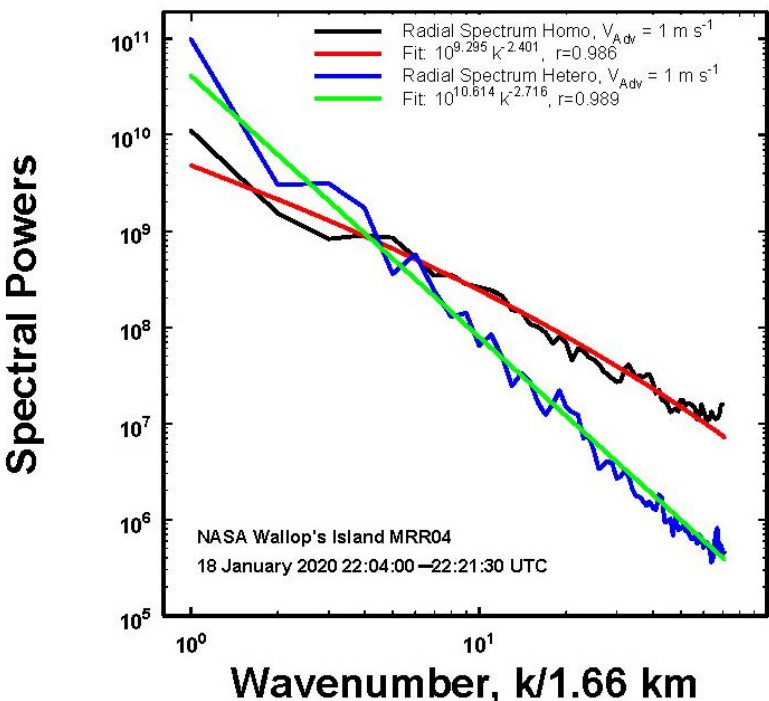

**Figure 12.** The computed radial power spectrum assuming an advection velocity of $1 \text{ m s}^{-1}$ for the period of steady rain and for the next 1060 s of statistically inhomogeneous, non-steady rain, as discussed further in the text.

Nevertheless, in this particular example, for a fixed mean rainfall rate of $50 \text{ mm h}^{-1}$ (such as what might be produced at a coarse, 5 km resolution by a numerical model or measured by a radar, which acts to filter out fine-scale structures [22,23]), we generated two synthetic timeseries of data from these different radial spectra fits. This was accomplished using the techniques of several different investigators (e.g., [24]) in which $L$-complex Fourier amplitudes ($A$) were created by assigning random phases ($\varphi$) to samples of the wavenumbers ($k$) at a fixed interval along $L$ from the fits. That is, the constructed series was

$$A_j = \sqrt{\frac{S_j(k)}{2L\Delta}} exp(i\varphi_k) \qquad (4)$$

where $S$ is the fit to the power spectrum. This series was then Fourier-transformed and complex-conjugated to obtain a data series consistent with the input spectral power fits.

These curves (Figure 13a) can be interpreted as observations by instruments over a 5 km area at one moment or by one instrument at one fixed location in time, where time is distance/$V_{Adv}$, and $V_{Adv}$ is the mean advection speed of the rain. Because of the similarity of the expressions for the spectral power fits in Figure 12, the structures in Figure 13 were

remarkably similar, but there were important differences in magnitudes, as reflected in the histograms (Figure 13b).

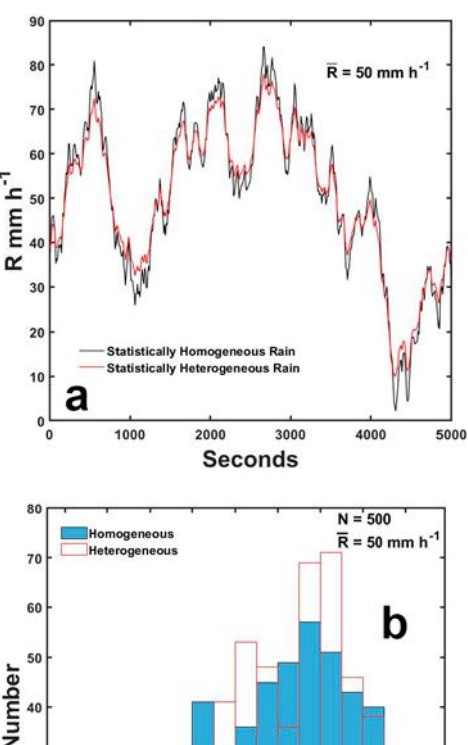

**Figure 13.** (**a**) Timeseries of synthesized data over 83 min using the two power spectra in Figure 12 showing real differences between the two; (**b**) histograms for each synthesized rainfall highlighting the differences in (**a**).

The maximum differences were about 10 mm h$^{-1}$ for the mean $R$ of 50 mm h$^{-1}$ or about 20% with an integrated total absolute difference of 1225 mm h$^{-1}$. While not huge, such differences could, at times, become significant, for example, when looking at storm run-off or at soil erosion.

While Figure 13a represents what might be seen in time (or a horizontal space of 5 km for $V_{adv} = 1$ m s$^{-1}$), Figure 14 shows what one realization might look like over a 2 km height not unlike what was observed in some of the data for height at one time presented here.

It also suggests how radar observations of rainfall rate might vary with altitude depending on the radar beam dimensions and geometry of the observations, as illustrated in Figure 15 for Marshall–Palmer rain [25] using the relation $Z = 200\,R^{1.67}$. The limit of the $x$-axis implied a variation in the radar reflectivity factor of 8 dBZ (a factor of 6) aside from the usual statistical signal fluctuations.

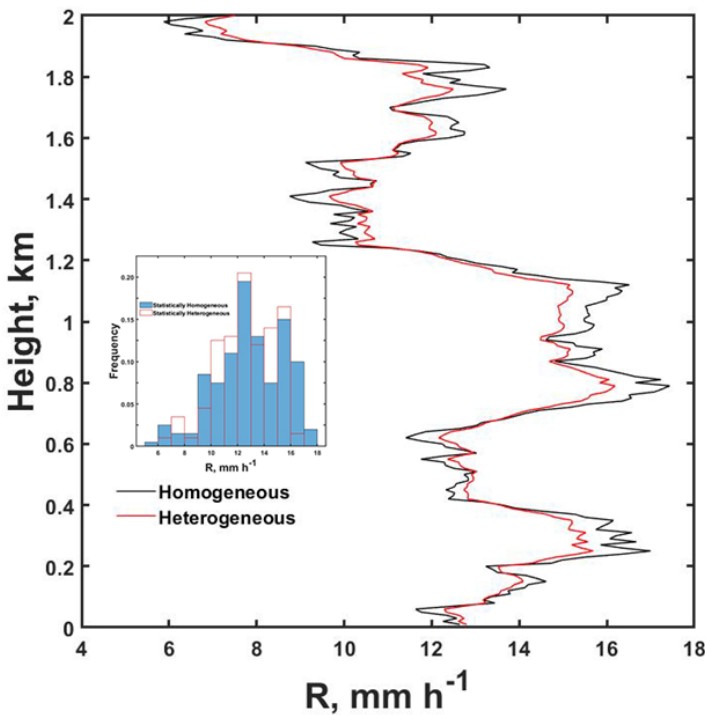

**Figure 14.** An example of the differences in the rainfall rates for statistically homogeneous and heterogeneous rain in height using the respective power spectral fits.

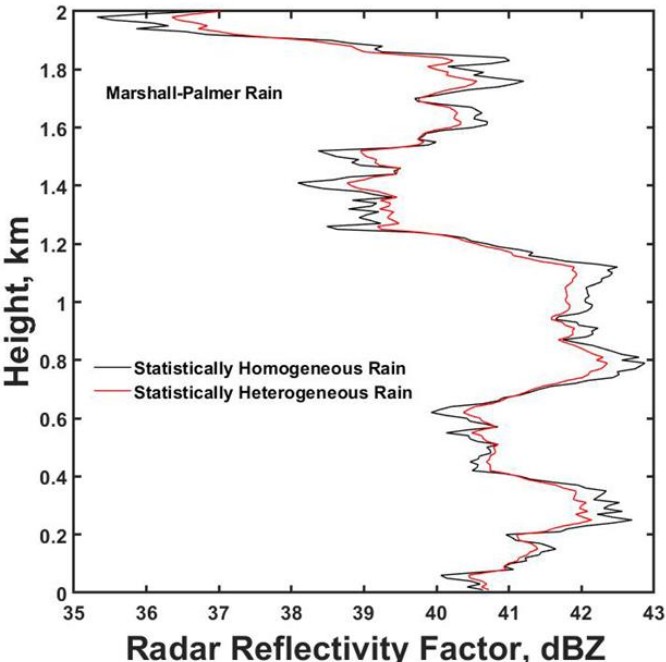

**Figure 15.** The radar reflectivity values corresponding to the assumption of Marshall–Palmer rain (Marshall and Palmer 1948) and corresponding to the synthesized rainfall rates in Figure 14.

### 3. Concluding Remarks

In this paper, fine-scale rainfall rates were calculated using Doppler radar measurements properly corrected for vertical air velocity and attenuation. This produced time–height profiles of rainfall rates that could be statistically analyzed over periods of varying length up to 1.28 km height at 10 m/10 s resolution for convective types of rain and for one case of less intense, steadier rain. With the exception of one period of steadier rain, the data were all found to be statistically heterogeneous with only very localized pockets of

statistically homogeneous rainfall. Consequently, in general, it was not possible to construct meaningful correlation functions or to use the Wiener-Khintchine theorem to transform such relations into radial power spectra. Instead, it was necessary to directly compute 2D power spectra using a Fourier transform after first assuming a fixed advection velocity of $1 \text{ ms}^{-1}$ so that the temporal axis could be transformed into a spatial axis These, in turn, were used to compute 1D radial power spectra applicable to each case only.

Nevertheless, in all cases, these radial power spectra could be well fit to the wave numbers by power relations with negative exponents ranging from 2.47 to 2.76 for both the NASA Wallop's Island MRR observations and those using the College of Charleston MRR radar over a year later at the different location that was near Charleston, South Carolina. The precise values, however, were shown to depend upon the assumed advection velocity so that, with greater advection speeds, the wavelengths were stretched, leading to larger exponents as discussed in the text. Consequently, the only way to obtain estimates of the true spectral exponents was to collect measurements using spatially separated radars—a process being undertaken within a current grant.

However, in spite of the limitations of the current data, useful conclusions are still possible. For example, based upon these observations and analyses, it appeared likely that convective rainfall data is predominantly statistically heterogeneous, so it will not be likely to ever have a 'universal' scaling relation for such rain. Consequently, the alternative is to collect such relations in different locations for different types of precipitation for use as references to better scale either radar rainfall estimates over larger beam dimensions above the surface or large-scale outputs from numerical forecast models for applications at smaller scales, such as those for rain run-off warnings or soil erosion research, for example. That is, as discussed above and illustrated in Figures 13–15, one can always synthesize Monte Carlo data using relations such as those in Figure 12 and then filter (for example, by averaging) the final rainfall series to match the scales of interest, as discussed in [3]. Additional general findings are also in process. In particular, this research is being extended to explore the behavior of radial power spectral fits in response to rainfall rates and to total spectral powers with some interesting findings appearing in a forthcoming paper currently under preparation.

**Author Contributions:** There are two authors who contributed to this work: A.R.J. and M.L.L. A.R.J. devised and developed this idea; M.L.L. provided some of the data associated with the College of Charleston, carefully checked the science, and made important suggestions. All authors have read and agreed to the published version of the manuscript.

**Funding:** This work was supported by the National Science Foundation (NSF) under grant AGS2001343 (A.R.J.) and by grants AGS2014900, 1823334, and 1532977 (M.L.L.).

**Institutional Review Board Statement:** Not applicable.

**Informed Consent Statement:** Not applicable.

**Data Availability Statement:** The data and Matlab programs are available at Jameson, Arthur (20222), 'MRR Data for Analyses', Mendeley Data, https://data.mendeley.com/datasets/rfw8h7h2pk/3, (accessed on 21 March 2022).

**Conflicts of Interest:** We wish to confirm that there are no known conflict of interest associated with this publication, and there was no significant financial support for this work that influenced its outcome. All of the sources of funding for the work described in this publication are acknowledged above.

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
