# Peer review of "Preliminary Statistical Characterizations of the Lowest Kilometer Time–Height Profiles of Rainfall Rate Using a Vertically Pointing Radar"

_atmosphere, doi:10.3390/atmos13040635_

Round 1

Reviewer 1 Report

The manuscript “Preliminary Statistical Characterizations of the Lowest Kilometer Time-Height Profiles of the Rainfall Rate Using a Vertically  Pointing Radar,” by Jameson and Larson presents analyses of time-height data acquired by high-resolution vertically-pointing radars.  They use analysis techniques previously used for measurements collected over time or at multiple horizontal locations.  This paper extends these techniques to data measured over altitude.  They find that for several cases of convective rainfall, with different radars and locations, the data are not statistically stationary except for generally very small areas.  They then go on to compute power spectra of the data and show how to use the results to re-construct data that are statistically stationary, or homogeneous.  I think this is an interesting study but needs some further clarification and revision prior to publication.  One area in which the paper could be improved is better motivation for the work.  Specifically, some examples of how the data might be used if stationary versus its use in light of the finding of being mostly heterogeneous.  Also, I have a number of questions on the validity of the wavenumber spectrum for data that isn’t stationary.  I think justification for the power spectra work could be improved.  Specific comments are below.

p. 2, line 61, while the definition of wide-sense stationary here is based only on mean and variance, the definition that I normally see is constant mean and autocorrelation independent of position; according to Papoulis’ book on Probabiity, RVs, and Stochastic Processes, the autocorrelation should depend only on the difference in space/time.

p. 2, line 70 – “minimum” => “minima”

p. 2, line 71, An explicit expression for alpha in the case of a rain record would be helpful. Alpha as used for temperature involves record highs and lows; is this the same for rainfall rate?

p. 3, line 96, Directional dependence could also be related to the isotropy of the random field. We could have an autocorrelation that is only a function of spatial difference but has different dependence in, say, x and y? -i.e., homogeneous but isotropic. I assume that the directional dependence noted here applies to determining the homogeneity but not necessarily isotropy?

p. 3, line 104, I can see this as a valid way but is it optimal?

p. 3, line 106, are the paths here continuous as in [14]; e.g, south, then north, then south?

p. 4, line 123, how does this analysis work? Even if described in another paper, a sentence or two would be useful to avoid having to go back to a reference.

p. 5, line 128, use reference number instead of authors.

p. 5, line 138, Is IXH computed using both directions?

p. 5, lines 144 and 145, should these sample numbers have another zero, i.e., 25000 and 26000?

p. 6, lines 161-162, is having such a small homogeneous area meaningful? 

p. 6, line 167, caption of Figure 4, it would be helpful to use an arrow to point out the homogeneous area.

p. 9, line 207, I think this needs more explanation. With no correlation functions, what is the definition of the spectrum? It seems like defining the spectrum for a non-stationary process is problematic. see R. M. Loynes, Journal of the Royal Statistical Society. Series B (Methodological) Vol. 30, No. 1 (1968), pp. 1-30. 

p. 9, line 210 and Figure 8, The results are plotted in time and height spectra.  Is the advection velocity used?  Both plots are in wavenumber; perhaps less confusing to use "vertical" and "horizontal" spectra.

p. 11, line 232, the verb seems to be missing in the sentence “The 2D horizontal-vertical coordinate system ...”.

p. 13, line 276, An explanation of "steady" versus "homogeneous" would be helpful.

p. 15, line 322, “two expressions” refers to the two fits in Fig. 12?

p. 15, line331, For a spaceborne radar in which the time-series in Fig. 12 could be considered rainfall across a footprint, the radar is simply averaging the antenna pattern with the rainfall reflectivity (ignoring effects of attenuation). I'm hoping the authors can comment on the assumptions for that versus Fig. 12.

p.16, line 342, Figure 15, how is the profile for Marshall-Palmer rainfall generated?

p. 17-18, lines 372-377: For the conclusion: “Consequently, rather than looking for universal scaling laws in such precipitation, it will likely be more productive to build a catalog ...” Both the basis for and application of this conclusion should be better explained.  I would recommend adding a paragraph or two to the previous section to better illustrate this point.  As is, the applications of the results may not be very obvious to many readers.  An example of how this would be used would be very helpful.

Reviewer 2 Report

The manuscript can be accepted in the present form.

Reviewer 3 Report

It's an excellent paper and such results deserve to be published. Some minor comments may be addressed.

  1. One of the issue is data length. The data length seems to be very small. Authors have showed a snap shots of one day. This may not bring out the statistics.
  2. How much uncertainty in the observation is estimated due to spatial heterogeneity?
  3. The uncertainties in the MRR data needs to be documented through some analyses.
  4. In case of heavy rain, what is the amount of attenuation in the MRR?

The paper is written very well and the results are interesting. I recommend to publish with minor revision
